# Ontogenetic Transfer of Microplastics in Bloodsucking Mosquitoes *Aedes aegypti* L. (Diptera: Culicidae) Is a Potential Pathway for Particle Distribution in the Environment

Anastasia Simakova *, Anna Varenitsina, Irina Babkina, Yulia Andreeva, Ruslan Bagirov, Vadim Yartsev and Yulia Frank *

Biological Institute, Tomsk State University, Lenina Ave. 36, 634050 Tomsk, Russia; annette.ander@yandex.ru (A.V.); bibsphera@gmail.com (I.B.); andreeva_y@mail2000.ru (Y.A.); rbaghirov631982@gmail.com (R.B.); vadim_yartsev@mail.ru (V.Y.)
* Correspondence: anastasiasimakova@yahoo.com (A.S.); yulia.a.frank@gmail.com (Y.F.)

**Abstract:** The uptake and accumulation of microplastics (MPs) by bloodsucking mosquitoes *Aedes aegypti* L., carriers of vector-borne diseases, were investigated in the laboratory. In the experimental group, polystyrene (PS) particles were registered in insects of all life stages from larvae to pupae and adults. *Ae. aegypti* larvae readily ingested MPs with food, accumulating on average $7.3 \times 10^6$ items per larva in three days. The content of PS microspheres significantly decreased in mosquitoes from the larval stage to the pupal stage and was passed to the adult stage from the pupal without significant loss. On average, 15.8 items were detected per pupa and 10.9 items per adult individual. The uptake of MPs by *Ae. aegypti* did not affect their survival, while the average body weight of mosquitoes of all life stages that consumed PS microspheres was higher than that of mosquitoes in the control groups. Our data confirmed that in insects with metamorphosis, MPs can pass from feeding larvae to nonfeeding pupae in aquatic ecosystems and, subsequently, to adults flying to land. Bloodsucking mosquitoes can participate in MP circulation in the environment.

**Keywords:** microplastics; PS microspheres; bloodsucking mosquitoes; ontogenetic transfer

## 1. Introduction

Plastic is a ubiquitous material in every aspect of human life, from food packaging and household products to high-tech medicine and electronics [1]. However, its widespread use, coupled with its low cost, has led to inefficient waste management and environmental pollution throughout the world [2,3]. An additional increase in plastic release into the environment due to the COVID-19 pandemic has been recorded [4–6]. Once in the environment, plastic items can fragment to form microplastics (MPs) of 1 μm to 5 mm and even smaller nanoplastics (NPs), which are <1 μm [7,8]. In addition, there is direct entry of microsized plastic into the environment with domestic and industrial waste [9]. The detection and study of the behavior of MPs in various environments and living organisms is a pressing area of research, also because of their adverse physiological effects in animals and humans [10,11].

Although MPs are detected in almost all environments, from soils and groundwater to the atmosphere, surface waters are one of the major sinks of MPs on the planet [12,13]. Many aquatic animals ingest MPs, mistaking them for food particles, or consume them passively, so that the plastic enters food webs [14–16]. Until recently, the bioaccumulation of MPs by aquatic organisms was known mainly for marine species. Currently, MPs have been detected in freshwater organisms of almost all trophic levels [17–19], and evidence has been obtained for their accumulation and biomagnification in the food webs of freshwater systems [20,21].

Mosquitoes (Diptera: Culicidae) are bloodsucking insects and carriers of vector-borne diseases in wild animals and in humans, including yellow fever, dengue, chikungunya, and Zika viruses [22–24]. Two species of mosquitoes, *Aedes aegypti* (Linnaeus, 1762) and *Aedes albopictus* (Skuse, 1894), have received much attention due to their significant role in the transmission of pathogens, for which *Ae. aegypti* is considered to be the primary vector [25]. Bloodsucking mosquitoes such as *Ae. aegypti* can participate in the circulation of MPs and NPs in the environment through the transfer of tiny particles in transmission networks of mosquito-vectored pathogens. Potential promotion of the (re-)emergence of infectious diseases by MPs has been suggested recently, including when transferred by insects [26]. In addition, mosquitoes are invertebrates that spend their juvenile life stages in water, but their flying adult stages on land, and therefore, they are able to carry MPs between aquatic and terrestrial environments.

The filter-feeding larvae of mosquitoes ingest latex microspheres, as was described for *Culex pipiens* L. [27] and *Ae. aegypti* L., *Anopheles albimanus* Wiedemann, *Anopheles quadrimaculatus* Say, and *Culex quinquefasciatus* Say [28]. More recently, evidence of MP ontogenic transference in *Culex* mosquitoes has been found [29]. *Culex* mosquito larvae ingested fluorescent polystyrene beads; 2 μm particles were readily transferred between life stages that use different habitats: from the larva into the pupa life stage and, subsequently, into the adult terrestrial stage [29,30]. Quantifying such transport would be very useful for modeling the processes of the global cycle of MPs.

The present study aimed at the quantitative assessment of ontogenetic transfer of fluorescent polystyrene MPs in amphibious mosquitoes *Aedes aegypti* along with the evaluation of the effect of MPs on the mass and mortality of the insects.

## 2. Materials and Methods

### 2.1. Preparation of MPs

Fluorescent yellow-green carboxylate-modified polystyrene $2.0 \pm 0.2$ μm microspheres (Sigma-Aldrich, St. Louis, MO, USA) with a density of 1.050 g cm$^{-3}$, excitation of 470 nm, and emission of 505 nm were used in all experiments. MPs were stored as a stock suspension (2.5 mg mL$^{-1}$) in distilled water and mixed using a vortex (Microspin BioSan FV-2400, Riga, Latvia) prior to dilutions. To break up the agglomerates and spread particles more evenly, MPs were washed before application by adding 1 mL from the stock solution into a 1.5 mL Eppendorf tube and then centrifuging at 9000 rpm for 10 min. The supernatant was discarded, and 1 mL of distilled water was added. The solution was then resuspended by using the vortex and centrifuged again at the same speed and duration. This process was repeated twice [29].

### 2.2. Mosquito Colonies

An *Aedes aegypti* mosquito colony is constantly maintained in the laboratory of evolutionary cytogenetics of Tomsk State University. Females are fed guinea pig blood twice a week. Cotton pads soaked in 10% honey solution were provided for additional sustenance of the insects. The larvae are kept in dechlorinated tap water. To feed the larvae, powder from dried beef liver is used with the addition of powdered dry *Urtica dioica* leaves. The colonies are maintained in the laboratory at $25 \pm 2$ °C, relative humidity $70 \pm 5\%$, and an alternating 16 h light and 8 h dark.

### 2.3. Experimental Protocols

The experiment was carried out in 5 parallel replications. In each replication, 15 larvae of the third instar *Ae. aegypti* were placed in a Petri dish ($90 \times 90$ mm) filled with 75 mL of tap water. The larvae of the experimental group were grown in the presence of PS microspheres (Section 2.1) by adding particles to water at a calculated concentration of $8.0 \times 10^6$ items mL$^{-1}$, as previously recommended [30]. The actual concentration of MPs was determined at the beginning and at the end of the experiment by taking 3 aliquots of 25 μL from different points of each dish and analyzing the number of particles microscopically. The larvae of the control

group were grown under similar conditions without the addition of plastic microspheres. The mortality of insects at each life stage was monitored and recorded daily during the entire period of the experiment.

In each replication, larvae were randomly taken after 3 days, pupae after 7 days, and adult individuals after emergence. The fourth instar larvae, pupae, and adult insects taken from the experimental and control groups were weighed using a microbalance with an accuracy of $\pm 0.001$ g (Jewelry scale 8068-series, Shenzhen, China). Adults were examined under an epifluorescent microscope to make sure that there were no MPs attached to the body surface. Then, all samples were washed twice with distilled water to remove MPs from the body surface of the mosquitoes and separately fixed with 70% ethanol in 1.5 mL tubes until the analysis.

For further microscopic analysis and quantification of MPs in insects, experimental and control mosquitoes in each replication were splashed with distilled water and homogenized individually with the addition of 35% $H_2O_2$ with 0.05M $FeSO_4$ as a catalyst in a ratio of 3/1 (v/v) for oxidative digestion until the tissues were completely dissolved. Wet peroxide oxidation is applied in different variations for extraction of MPs from natural water, sediments, and tissues of fish and invertebrates [31–33].

The homogenate from each larva was carefully shaken, and 2 $\mu$L of the liquid was randomly selected in triplicate from each tube. Micropreparations were prepared from subsamples and examined using epifluorescent microscopy (Section 2.5). MPs were counted at least in 30 fields for each subsample and averaged. The content of MPs in each studied individual was determined in this way. The results are expressed in MP items per mL of homogenate and recalculated for each inspected individual based on the final volume of the sample after homogenization and wet peroxide oxidation. To transfer the particle number per individual into mass concentration, the first one was divided by the mass of the individual. The homogenate from pupae and adults was analyzed the same way, but the entire volume of the obtained homogenate was analyzed and the number of MPs in each pupa and adult was counted.

### 2.4. Histological Observations

Histological thin sections were prepared as described earlier [34] using the fourth instar larvae from the experimental group. Larvae were fixed for 24 h in 10% formalin solution buffered with cacodylate buffer, pH 7.4. Then, the studied insects were washed with distilled water and dehydrated in a graded series of ethanol solutions (70%, 95%, absolute EtOH), cleared in 100% butanol, soaked, and embedded in paraffin (LabPoint, Russia). Serial sagittal sections (thickness 5 $\mu$m) were prepared using a rotary microtome RMD-3000 (MTPoint, Saint Petersburg, Russia). Sections were fixed on SuperFrost slides and stored before microscopic analysis. Epifluorescence microscopy was utilized for microscopic analysis and imaging, as described in Section 2.5.

### 2.5. Microscopy and Imaging Techniques

Epifluorescence microscopy (Axio Imager Z1, Carl Zeiss, Oberkochen, Germany) equipped with the filter set for FITC fluorescein was used for counting and imaging of PS microspheres, magnification $\times 400$. AxioCam MRm (Carl Zeiss, Germany) and AxioVision Rel. 4.7 software was utilized to obtain microphotographs.

Images of *Ae. aegypti* larvae and pupae were obtained using a stereomicroscope Stemi 200-C equipped with AxioCam ERs 5 s digital camera and Zen software.

### 2.6. Statistical Methods

The data were analyzed using the statistical software R v4.0.5 [35]. The effects of MP treatments on adult weights were analyzed using ANOVA following log10 transformation to meet normality and for the homogeneity of variance across groups (Shapiro–Wilk test, $p > 0.05$; Levene's test, $p > 0.05$). We performed post hoc Tukey's comparisons where terms significantly affected a response variable at the 95% confidence level [36].

The number of MPs at the stages of larva, pupa, and adult mosquitoes was estimated separately as described above (Section 2.3). Comparison of MP accumulation in different stages of mosquitoes was evaluated using the Kruskal–Wallis test [37], $p < 0.05$. To assess differences in the mortality rate, Fisher's exact test [38] was applied, $p < 0.05$.

## 3. Results

A high initial concentration of MPs was created in the experiment, which was $8.0 \times 10^6$ items $mL^{-1}$. Microscopic analysis of the experimental medium showed that the actual initial concentration of particles corresponded to the calculated concentration. As a result of the experiment, a decrease in the initial concentration of MPs by 2.4 times was observed (data not shown). The decreasing content of MPs in the medium was connected to active ingestion of particles by insects. Larvae mainly consumed PS particles (Figure 1). As can be seen in the histological sections (Figure 1b), MP particles were localized mainly in the intestines of the larvae.

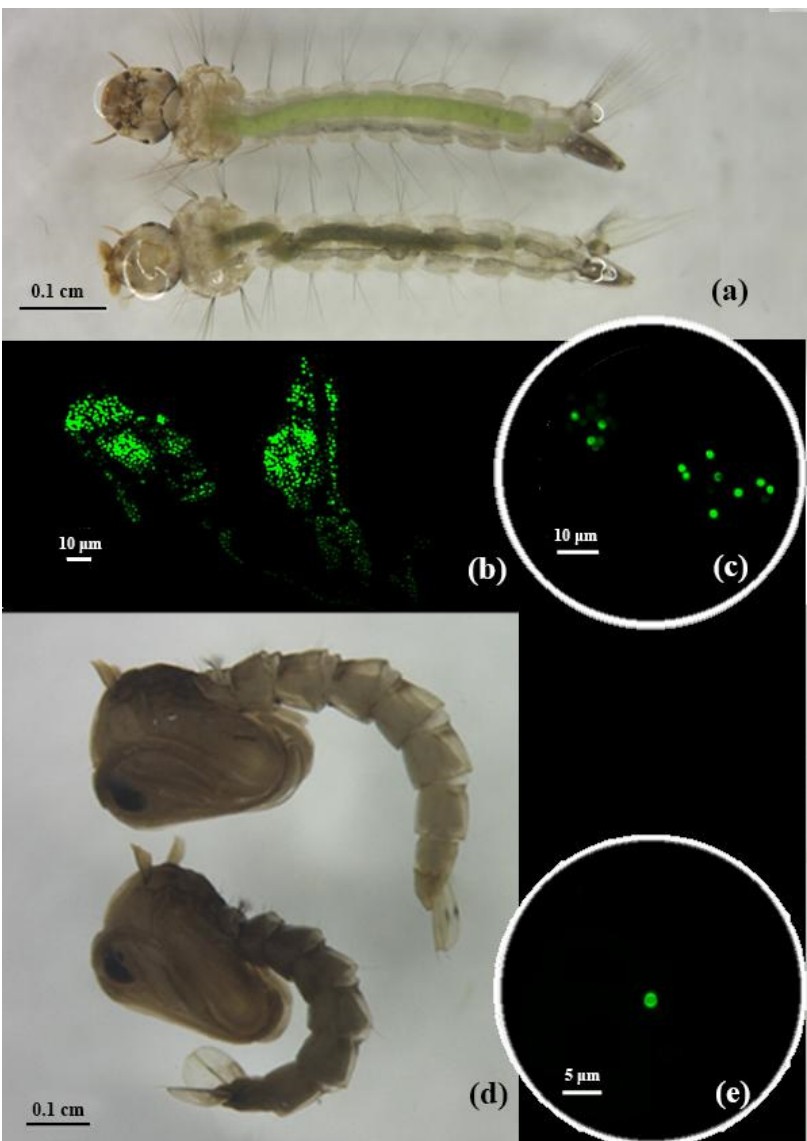

**Figure 1.** Habitus of *Ae. aegypti* larvae from the experimental group on top (intestines filled with MPs), larva from the control group on the bottom, light microscopy (**a**); thin section of the intestine of the larvae from the experimental group (**b**) and microspheres detected in larvae (**c**), epifluorescent microscopy; habitus of *Ae. aegypti* pupa from the experimental group on top, pupa from the control group on bottom, light microscopy (**d**); microspheres detected in pupa, epifluorescent microscopy (**e**).

In mosquito larvae, the average content of PS microspheres was $64.0 \times 10^6$ MPs per mL of homogenate. One larva ingested on average $7.30 \times 10^6$ items for 3 days (Figure 2a). No MPs were found in control group replicates.

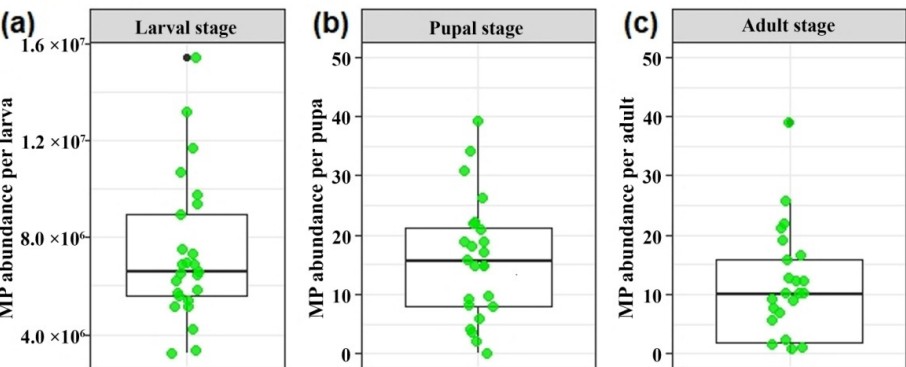

**Figure 2.** Abundance of MPs in mosquitoes at different life stages expressed as number of MPs per larva (**a**), per pupa (**b**), and per adult individual (**c**). The green circles show the distribution of the studied individuals. Each box includes three horizontal lines, which denote 25%, 50% (median), and 75% of the data (from 25% to 75% of the data are enclosed in a rectangle); the upper whisker extends from the first quartile to the maximum value no further than $1.5 \times$ IQR (where IQR is the interquartile range); lower from the third quartile to the smallest value, no further than $1.5 \times$ IQR of data; (●), values beyond $\pm 1.5 \times$ IQR.

Ontogenetic MP transference from larvae to pupae and adult mosquitoes was confirmed by fluorescent microscopy and quantified (Figures 1 and 2). Quantitatively, the ontogenetic transfer of microplastic particles into pupae and adults was relatively low. Two-micrometer PS spheres in a small amount emerged in the pupal stage and without significant losses into the adult stage. If there were $7.30 \times 10^6$ items per larva, then the number of particles per pupa averaged 15.8 MPs (Figure 2b). In adult mosquitoes, a generally insignificant ($p > 0.05$) decrease in the number of particles to 10.9 MPs per individual was observed (Figure 2c). The ratio of mosquitoes that did not show MPs in pupae and adults did not differ. The percentage of individuals in which ontogenetic transfer was detected was 92%, and that in which transfer was not detected was 8%.

The survival rate of mosquitoes at all life stages remained quite high (Figure 3). The highest mortality was detected at the larval stage, 24% in the control and 28% in the experimental group. At the pupal and adult stages, mortality did not exceed 8%. There were no statistically significant differences in survival between control and experiment groups (Fisher test, $p = 0.61$) (Figure 3). Extremely high concentrations of MPs used in the experiment did not affect the viability of *Ae. aegypti*.

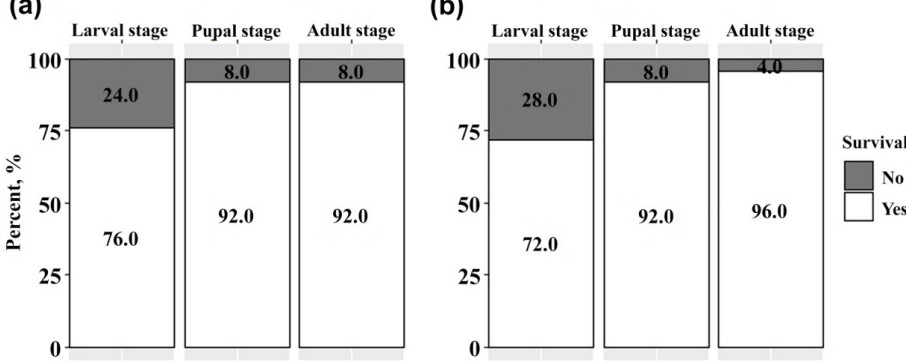

**Figure 3.** Mortality and survival of mosquitoes at different stages of the life cycle in control (**a**) and experimental conditions (**b**).

In addition to survival, the effect of MPs on the body weight of mosquitoes was evaluated. The average body weight of insects at all life stages in the control was less compared to the experimental group, $1.20 \pm 0.07$ mg and $1.80 \pm 0.12$ mg, respectively (Table 1).

**Table 1.** Average body weight of mosquitoes at different life stages in control and experimental conditions (mean $\pm$ standard deviation).

| | Body Weight, mg | | | |
|---|---|---|---|---|
| Condition | Larvae | Pupae | Adults | Average across Stages |
| Experiment | $1.60 \pm 0.22$ | $2.30 \pm 0.22$ | $1.40 \pm 0.15$ | $1.80 \pm 0.12$ |
| Control | $1.10 \pm 0.08$ | $1.60 \pm 0.14$ | $0.90 \pm 0.07$ | $1.20 \pm 0.07$ |

The average weight of larvae in the control and in the experiment was slightly less than that of pupae (Figure 4). The average weight of adult mosquitoes in the control and in the experiment was significantly less than that of pupae (1.60 mg and 2.30 mg, respectively) and insignificantly less than that of larvae (Figure 4).

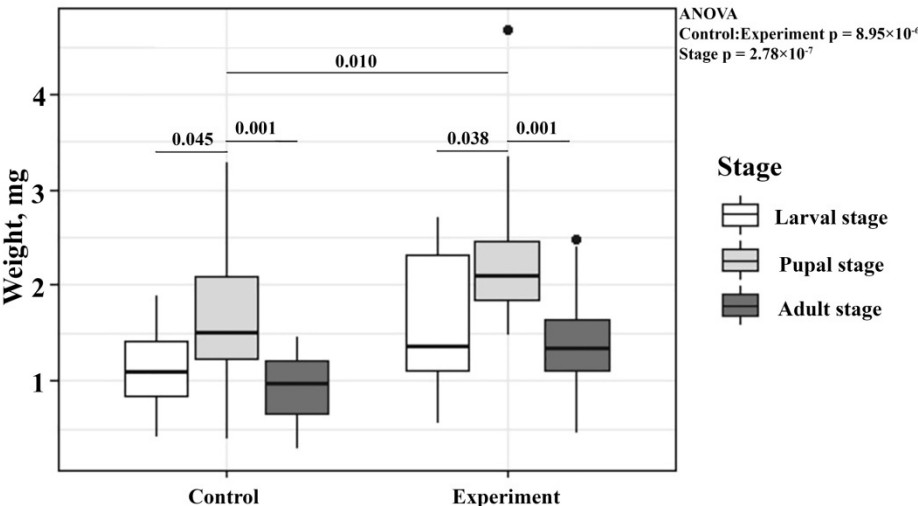

**Figure 4.** Individual body weight of mosquitoes at different life stages in control and experimental conditions. Each box includes three horizontal lines, which denote 25%, 50% (median), and 75% of the data (from 25% to 75% of the data are enclosed in a rectangle); the upper whisker extends from the first quartile to the maximum value no further than $1.5 \times$ IQR (where IQR is the interquartile range); lower from the third quartile to the smallest value, no further than $1.5 \times$ IQR of data; ($\bullet$), values beyond $\pm 1.5 \times$ IQR.

Thus, the mass of mosquitoes at all life stages was higher under the experimental conditions than in the control. Extremely high concentrations of MPs modeled in the experiment influenced the body weight of *Ae. aegypti* mosquitoes.

## 4. Discussion

MPs enter aquatic food chains, which is confirmed by the detection of particles in living organisms at almost all trophic levels. Freshwater invertebrates and fish, as well as marine life, consume MPs with food or while breathing [39]. MPs can be transferred from one trophic level to higher levels in marine and freshwater trophic chains [40–42].

Insects are the most numerous and diverse group of organisms on the planet [43,44]. The ubiquitous nature of MP pollution makes contact between plastics and insects inevitable along with the participation of the insects in particle trophic transfer. However, little is known about how this contact affects the basic ecosystem functions of insects [45].

Freshwater insects of different feeding groups ingest MPs in natural ecosystems [46,47]. Approximately 50% of benthic insects ingest MPs in riverine bottom sediments indepen-

dently of feeding guild and biological traits [48]. As was found in laboratory experiments, aquatic larvae of different mosquito species (*Aedes aegypti* Linnaeus, *Anopheles albimanus* Wiedemann, *Anopheles quadrimaculatus* Say, *Culex quinquefasciatus* Say) readily consume MPs, but the first instar larvae are unable to ingest latex spheres up to 45 μm in diameter [27,28]. Particle size influenced ingestion of PS spheres by other freshwater invertebrate species [49]. It has been previously established that the size of MPs is a very important factor in their uptake and ontogenetic transfer. Small MPs can be accumulated and transferred faster than larger ones. Two-micrometer PS spheres are better involved in ontogenetic transfer in *Culex pipiens* mosquitoes [29,30]. Al-Jaibachi and coauthors established the dependence of the ingestion of MPs on the concentration and size of particles. A greater number of particles was consumed by insects where the concentration of MPs was higher and the particle sizes were smaller. MPs of 2 μm were able to transfer from larvae into adults much more often than 15 μm particles, which is probably related to the accumulation of MPs in the Malpighian tubes [29,30]. Thus, it was not by chance that we chose small (2-μm) MPs in an extremely high concentration of $8.0 \times 10^6$ items mL$^{-1}$.

Our study confirmed previous data on the ontogenetic transfer of MPs in amphibious insects, using mosquitoes. It was shown in this study that MPs are readily ingested by *Ae. aegypti* larvae with food, partially pass into the nonfeeding aquatic pupae, and then pass into the flying adults, which live in the terrestrial environment. During the transition from the larval stage to the pupal, significant losses of MPs were observed, while these losses were insignificant during the transition from the pupa to the adult mosquito (on average from $7.30 \times 10^6$ items per larva to 15.8 items per pupa and to 10.9 items per adult individual).

In earlier studies [29,30], at a maximum 2 μm MP concentration, 3,4-fold losses of particles were observed in *C. pipiens* during the transition from larvae to pupae. While transitioning from the pupa to the adult stage, the losses were significantly greater: 22–24-fold. On the contrary, in our study, the loss in the number of MPs during the transition from *Ae. aegypti* larvae to pupae was colossal ($4.6 \times 10^5$-times), while from pupae to imagoes (adults), it was insignificant (only 1.5-times). Small MPs particles trapped in pupae were almost entirely carried from the aquatic environment into the air and terrestrial sites by adult flying mosquitoes. Ontogenetic transfer may vary among different genera.

We found that such a high PS microsphere concentration as $8.0 \times 10^6$ items mL$^{-1}$ had no effect on the mortality of larvae, pupae, and adults of *Ae. aegypti*. The survival rate of insects at all life stages was very high. There was also no evidence that lower concentrations of MPs had any significant effect on the survival of *C. pipiens* larvae up to the terrestrial adult stage [25,29], although other organisms and other types of MPs have shown toxic effects for aquatic insects. Polyethylene (PE) particles from 1–4 μm to 100–126 μm at relatively low, environmentally relevant concentrations (500 items kg$^{-1}$ bottom sediments) negatively affected the survival, growth, and emergence of *Chironomus tepperi* [50]. Exposure of *Chironomus riparius* larvae to irregularly shaped PE microparticles at higher concentrations (from 1.25 g kg$^{-1}$ to 20.0 g kg$^{-1}$ sediment) led to a significant reduction of similar magnitude in larval growth and a significant delay in the emergence of imagoes [51].

MP consumption in the aquatic environment can affect the feeding behavior of hydrobionts, but the available data are controversial [17,52]. To assess such effects, the influence of MPs on the body weight of animals can be analyzed. Previously, no negative effect of PS microspheres at concentrations up to $8.0 \times 10^5$ on the weight of *C. pipiens* adults was shown, which indicated that the larvae did not suffer from a lack of nutrition during their development [29]. In this study, we found that higher concentrations of PS spheres ($8.0 \times 10^6$) affected the feeding behavior of *Ae. aegypti* larvae. The presence of MPs significantly affected the average body weight of mosquitoes at larval, pupal, and adult stages. The most likely scenario is this: larvae readily ingest MPs, but the body is not saturated with nutrients. This is the probable reason for the larvae beginning to feed more actively in preparation for metamorphosis. That is why the experimental mosquitoes gained more weight than the control group. The energy spent on metamorphosis in control and exper-

imental insects is comparable. In this regard, the nonfeeding stage pupae and then the adults also exceed the control groups of mosquitoes in average body weight. Similarly, in *D. magna* exposed to MPs, a positive growth response was observed at a high food (algae) concentration with up to 54% increase; gut passage time doubled for daphnids exposed to high concentration of some MP types [53].

Thus, taking into account the absence of the influence of high MP concentrations on the development and mortality of mosquitoes, it is likely that mosquitoes that uptake MPs in aquatic ecosystems will subsequently spread particles through the air into terrestrial food webs. The efficiency of MP transfer by bloodsucking mosquitoes depends on both the concentration and size of MPs [29], with significant losses from larval to pupal and insignificant losses from pupal to adult stages.

## 5. Conclusions

The study of the bloodsucking mosquitoes *Aedes aegypti*, which are important carriers of dangerous transmissible animal and human diseases, demonstrated in the laboratory that in these insects with complete metamorphosis, MPs can pass from feeding aquatic larvae to nonfeeding pupae and adults that fly to land. Two-micrometer PS microspheres were readily ingested by larvae, affecting the feeding behavior of the larvae and with an increase in body weight without affecting the development and mortality of the mosquitoes. Thus, bloodsucking mosquitoes may participate in the circulation of MPs, carrying particles from aquatic to terrestrial environments.

As a next step, experiments with other types and sizes of MPs are required. Future histological study of the possibility of MP bioaccumulation in other organs and tissues of larvae apart from the intestine would be appropriate because the question of the possibility for MP bioaccumulation in organs and tissues of larvae remains open. The study of MP abundance in larvae, pupae, and adults of *Aedes* mosquitoes from natural populations will also contribute to the understanding of MP transfer by bloodsucking insects.

**Author Contributions:** A.S. and Y.F. conceived of the study and contributed to the design and implementation of the research; A.V., Y.A. and R.B. performed the experimental work; I.B., V.Y. and A.S. contributed to the laboratory work and the analysis of the results; A.S. and Y.F. wrote the manuscript. All authors have read and agreed to the published version of the manuscript.

**Funding:** This research was supported by the Tomsk State University Development Programme (Priority2030).

**Institutional Review Board Statement:** Not applicable.

**Informed Consent Statement:** Not applicable.

**Data Availability Statement:** Not applicable.

**Acknowledgments:** We would like to express our great appreciation to Jean Kollantai, MSW (Tomsk State University) for the insightful comments and valuable suggestions on the style, which helped us significantly improve the paper.

**Conflicts of Interest:** The authors declare no conflict of interest.

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
