# Peer review of "Ontogenetic Transfer of Microplastics in Bloodsucking Mosquitoes Aedes aegypti L. (Diptera: Culicidae) Is a Potential Pathway for Particle Distribution in the Environment"

_water, doi:10.3390/w14121852_

Round 1

Reviewer 1 Report

The paper entitled `Ontogenetic transfer of microplastics in bloodsucking mosquitoes Aedes aegypti L. (Diptera: Culicidae) is a potential pathway for particle distribution in the environment` presents an interesting approach on the transfer of microplactics that can contribute to the literature. However, there are some questions that should be answered before being accepted for publication.

Introduction:

Line 37: There are also studies that show soil systems may be the main sink for microplastics. I`d recommend removing the certain statement `surface waters are the main sink of MPs on the planet`, or replacing that they can be one of the major sinks.

Material and Methods:

Line 73: What is the reason for particle washing? Explain.

Line 75: May there be a loss of microplastics during the centrifuging (e.g. sticking the walls of the tube, or when replacing the supernatant with the distilled water)

Line 91: Please give a brief description of the calculation.

Line 96: How the particle number was transferred into mass concentration, please explain.

Line 185: 24 and 28% should have the same sign (e.g `minus`).

Why mortality is higher during the larval stage?

Figure 4. Specify where the significant differences occur? (The significant difference signs are generated compared to the control group, or within the group?) If possible, better to add `letter signs` to show which one differs from the others to make it more clear.

Conclusion.

Future research recommendations could be added.

Author Response

Dear Reviewer,

Thank you for the appreciation of the article and for the useful review.

We made following changes in the text and gave some explanations in response to your comments.

Introduction:

Line 37: There are also studies that show soil systems may be the main sink for microplastics. I`d recommend removing the certain statement `surface waters are the main sink of MPs on the planet`, or replacing that they can be one of the major sinks.

We agree. The statement was corrected.

Material and Methods:

Line 73: What is the reason for particle washing? Explain.

Since the microplastic particles agglomerate during storage, they were washed and centrifuged to break up the agglomerates and spread more evenly.

An explanation added to text.

Line 75: May there be a loss of microplastics during the centrifuging (e.g. sticking to the walls of the tube, or when replacing the supernatant with the distilled water)?

We cannot exclude some losses, but they were insignificant. The concentration before and after these actions was controlled and it stayed unchanged.

Line 91: Please give a brief description of the calculation.

A brief description added in section 2.3.

Line 96: How the particle number was transferred into mass concentration, please explain.

A brief explanation was added in section 2.3.

Line 185: 24 and 28% should have the same sign (e.g `minus`).

Corrected.

Why mortality is higher during the larval stage?

These are random factors; perhaps during the transplantation of larvae for the experiment, some individuals were slightly injured. In general, mortality at all stages was very low.

Statistical differences in the mortality of larvae, pupae, and adults were insignificant (Fisher test, p=0.61). This is indicated in the text.

 Figure 4. Specify where the significant differences occur (The significant difference signs are generated compared to the control group, or within the group?) If possible, better to add `letter signs` to show which one differs from the others to make it more clear.

Figure 4 was corrected according to the comment.

Conclusion.

Future research recommendations could be added.

Future research recommendations previously given in the Discussion section were moved to the Conclusion and supplemented.

Reviewer 2 Report

I like the paper, it is well written and even after a careful reading I am not able to identify typo or mistakes. Literature is scarce on the topic, which makes the paper interesting.

Author Response

Dear Reviewer,

Thank you so much for the kind review.

Reviewer 3 Report

The article deals with the study of the uptake and accumulation of microplastics by bloodsucking mosquitoes Aedes aegypti L. The microplastics used were fluorescent polystyrene (PS). The article is well written and the results are clear. There is a huge amount of papers about microplastics, but most of them are related to the marine environment or freshwater, while the effect on insects and consequently on other systems like soil is less discussed. There are other papers in literature about the ontogenetic transfer of microplastics in mosquitoes, thus the novelty of the paper is not high. Anyway, I have appreciated that in the discussion part the data has been compared with the most relevant literature. Most of the results obtained in this paper are in line with the papers cited.

My comments:

Missing experiment

-Change the size of microplastics, to find a threshold value between 2 micrometers and 15 micrometers (these values are reported in the literature, but we don't know the effects in this range).

-In the abstract it is written items per imago individual. I don't know the public of this journal, but imago cannot be clear as term, so maybe in the abstract it can be written as adult and in the main text it can be explained that imago is the last part of the metamorphosis. 

-Some other references:

https://doi.org/10.1016/j.tifs.2021.05.027 (add after COVID-19 pandemic has been recorded)

https://doi.org/10.1021/acs.est.1c01753 and https://doi.org/10.3390/toxics9090224 (add after food webs)

https://doi.org/10.1016/j.envpol.2020.115750 (add after biomagnification in the food webs of fresh-water systems)

-Line 116 replace ml with mL

-I would recommend to use MP+ name (example MP ingestion) for the genitive and MPs when microplastic is plural (example ingestion of microplastics MPs).

Author Response

Dear Reviewer,

Thank you for the comments that helped to improve the article. We made changes in the text as follows:

Missing experiment

-Change the size of microplastics, to find a threshold value between 2 micrometers and 15 micrometers (these values are reported in the literature, but we don't know the effects in this range).

We plan to use 15 µm and, probably, other sized microparticles of different polymers in the future experiments to find threshold values. The information was added in the Conclusion that as a next step, experiments with other types and sizes of MPs are required.

-In the abstract it is written items per imago individual. I don't know the public of this journal, but imago cannot be clear as term, so maybe in the abstract it can be written as adult and in the main text it can be explained that imago is the last part of the metamorphosis.

Agree. “Imago” in the abstract changed to “adult” to make it more clear for a wide range of readers.

-Some other references:

https://doi.org/10.1016/j.tifs.2021.05.027 (add after COVID-19 pandemic has been recorded)

https://doi.org/10.1021/acs.est.1c01753 and https://doi.org/10.3390/toxics9090224 (add after food webs)

https://doi.org/10.1016/j.envpol.2020.115750 (add after biomagnification in the food webs of fresh-water systems)

Added.

-Line 116 replace ml with mL

Corrected.

-I would recommend to use MP+ name (example MP ingestion) for the genitive and MPs when microplastic is plural (example ingestion of microplastics MPs).

Thank you for the helpful suggestion, and we have made several corrections accordingly.

Round 2

Reviewer 3 Report

The paper is ready to be published after some minor changes:

-"To break up the agglomerates and spread particles more evenly were washed before application by adding..." Add a subject before were washed.

-"while from pupae to imagoes". The first time you name imago add between brackets adult.

In the supplementary file section I found the Word version of the article and not the supplementary data. Fix this issue.

Author Response

Dear Reviewer,

Thank you for the useful review.

We made following changes:

  • "To break up the agglomerates and spread particles more evenly were washed before application by adding..." Add a subject before were washed.

Added.

  • “while from pupae to imagoes”. The first time you name imago add between brackets adult.

Done.

  • In the supplementary file section I found the Word version of the article and not the supplementary data. Fix this issue.

Fixed.